# On a New Type of Combined Solar–Thermal/Cold Gas Propulsion System Used for LEO Satellite's Attitude Control

**Constantin Sandu [1], Valentin Silivestru [1], Grigore Cican [2,*], Horațiu Șerbescu [1], Traian Tipa [1], Andrei Totu [1] and Andrei Radu [1]**

[1] National Research and Development Institute for Gas Turbines COMOTI, 220D Iuliu Maniu, 061126 Bucharest, Romania; constantin.sandu@comoti.ro (C.S.); valentin.silivestru@comoti.ro (V.S.); horatiu.serbescu@comoti.ro (H.Ș.); traian.tipa@comoti.ro (T.T.); andrei.totu@comoti.ro (A.T.); andrei.radu@comoti.ro (A.R.)

[2] Faculty of Aerospace Engineering, Politehnica University of Bucharest, 1-7 Polizu Street, 011061 Bucharest, Romania

\* Correspondence: grigore.cican@upb.ro

**Abstract:** This paper presents the development, construction and testing of a new type of solar–thermal propulsion system which can be used for low earth orbit (LEO) satellites. Currently, the vast majority of LEO satellites are fitted with a cold gas propulsion system. Although such a propulsion system is preferred, the service duration of an LEO satellite is limited by the amount of cold gas they carry onboard. In the case of the new type of solar–thermal propulsion system proposed in this paper, the cold gas is first transferred from the main tank in a cylindrical service tank/buffer tank which is placed in the focal line of a concave mirror. After the gas is heated by the solar light focused on the service tank by the concave mirror, it expands by opening the appropriate solenoid valve for the satellite's attitude control. In this way the service duration of LEO satellite on orbit can increase by 2.5 times compared with a classic cold gas propulsion system. This is due to the propellant's internal energy increase by the focused solar light. This paper also presents the results achieved by carrying out tests for the hot gas propulsion system in a controlled environment.

**Keywords:** solar energy; solar–thermal; cold gas; propulsion system; LEO satellites

## 1. Introduction

Since the first satellite, Sputnik, was launched in 1957, more than 5560 successful launched operations occurred, which means about 9600 satellites in their orbits around the Earth's.

The European Space Agency (ESA)'s Space Debris Office estimated that about 5500 of these satellites were in space until February 2020, while 2300 were still functioning [1], most of which are low earth orbit (LEO) satellites. These satellites operate fairly close to the surface of the Earth, between 500 and 2000 km.

Because the number of artificial satellites in the Earth's orbit is increasing, their life-time must increase so that the quantity of space debris will not increase in a way that will affect space security, i.e., collision between satellites or spacecraft [2]. A satellite's operational life is measured also taking into account the quantity of propellant taken onboard and utilized for the orbit-position correction system, attitude control and specific maneuvers on the orbit. The efficiency of these correction systems depends on the propulsion system type. The most common types are cold gas propulsion systems, chemical propulsion, electric propulsion, propellantless propulsion systems—solar sails and solar thermal propulsion, etc.

More details about this propulsion systems can be found in [3–5].

There is an increasing effort to develop correction systems with an increased efficiency by utilizing energy outside the satellite, e.g., the solar energy for solar sails [6,7] or solar energy for solar–thermal propulsion [8].

Furthermore, the operational life increase of the satellite leads to environmental protection, the saving of energy, materials, manpower and finally, cost saving.

The use of solar energy in the cosmic space was imagined for propulsion systems fitted with a mirror [9] but also for other activities, e.g., space solar power satellite [10] or a specific surface illumination [11].

The idea of a mirror in space to reflect sunlight and thus generate power and light on Earth was proposed in 1928 by Hermann Oberth who postulated a space-manufactured 5 mm-thick mirror using sodium for the reflective layer, orbiting Earth at a 1000–5000 km altitude [12]. Across time, various solar concentration schemes have been proposed, including lenses, deployable arrays and inflatable arrays [13–15].

A very comprehensive conceptual, technical and socio-economic study and the exhibition of space mirrors was conducted by the space visionary Krafft Ehricke [16]. He proposed and analyzed in some detail a number of generic applications for providing lunar-type night illumination service ("Lunetta"), solar-type light energy services ("Soletta"), insolation for bio-production enhancement ("Biosoletta") to produce food and biomass, insolation for agricultural weather stabilization, precipitation management, crop drying and desalination ("Agrisoletta") and insolation for generating electricity on earth ("Powersoletta").

A significant brief step in the development of space mirrors was the Russian Space Mirror Project "Znamya" (banner) developed by the "Space Regatta Consortium" (SRC) established in 1990 by the Russian Space Agency and the corporation Energia [17]. Detailed information about the Znamya experiments is somewhat sparse [18].

Research on solar–thermal propulsion has been conducted for more than 60 years [19], yet no flight testing has been achieved. An overview of the milestones and program associated with solar–thermal technology since the advent of the technology in 1956 is provided [20].

The solar–thermal propulsion concept is a fairly basic one, relying on highly concentrated sunlight to heat a monopropellant to very high temperatures (typically approaching 3000 K) and subsequently exhausting the propellant through a discharge nozzle to provide thrust.

A typical solar–thermal engine is composed of three primary components or subassemblies:

- A solar concentrator, which concentrates and focuses solar energy onto the receiver;
- The receiver itself, which acts as a heat exchanger for the propellant;
- The propellant storage and feed system, which includes tankage and feed lines for routing the propellant to the receiver.
- A fourth component, control electronics, is not considered here but it is required to retrieve system telemetry, open and close valves, and (potentially) control both the concentrator and receiver position.

A working principle scheme of a solar–thermal propulsion system [21], further called STP, is presented in Figure 1.

A benefit of using STP as a spacecraft propulsion system is that it uses solar radiation directly to heat the propellant, therefore improving the conversion efficiency over resisto-jets that require converting solar radiation into electrical energy and then into thermal energy to heat the propellant. The operating temperature is limited by the materials available.

A cold gas propulsion system (CGP) relies on the process of controlled ejection of compressed liquid or gaseous propellants to generate thrust. Due to the absence of a combustion process, a CGP system requires only one propellant (without an oxidizer), and hence can be designed with minimal

complexity. The schematic of a typical CGP system is shown in Figure 2, and the main components include a propellant storage and a nozzle [22].

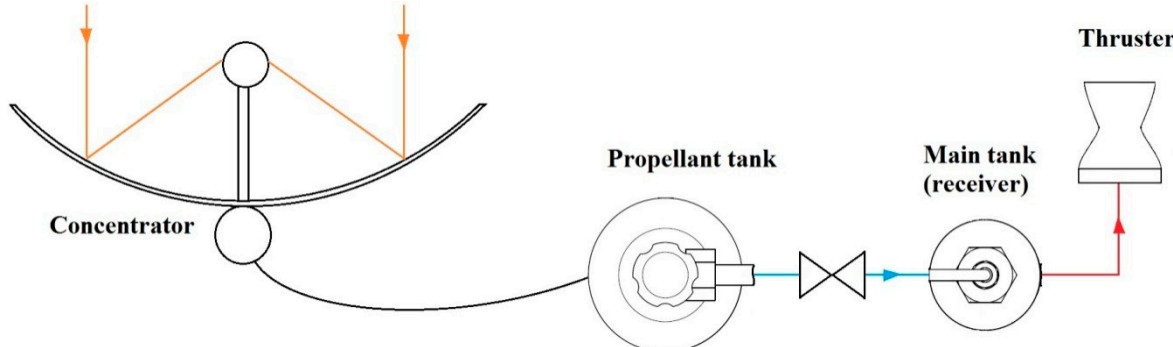

**Figure 1.** Schematic of a solar thermal propulsion system.

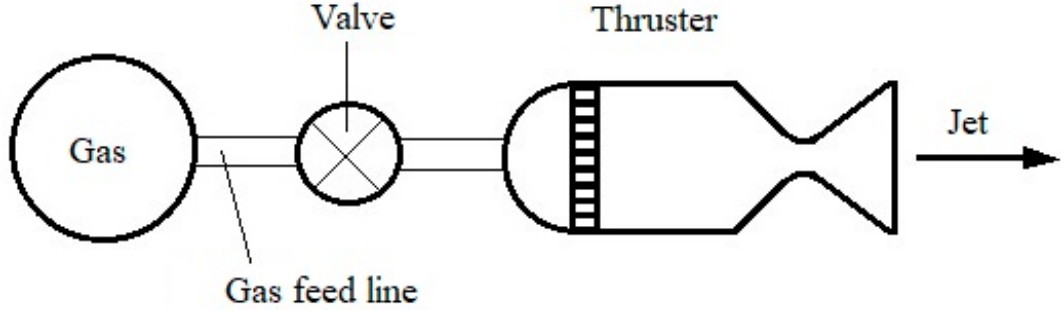

**Figure 2.** Schematic of the cold gas propulsion system.

The simpler design of a CGP system leads to a smaller system mass and lower power requirements for regulation purposes. However, these advantages come at the cost of a monotonically decreasing thrust profile over a period of time. The thrust produced is directly proportional to the pressure of the propellant inside the tank (propellant storage) and over the course of the mission, the tank pressure decreases (due to propellant usage) resulting in a decrease in the maximum thrust generated by the system. There are many studies and applications about cold gas systems, CGPs [23–26].

The novelty element of the present paper consists in the fact that it proposes a solution which combines the solar–thermal propulsion and cold gas propulsion concept, so that the solar energy which is concentrated from a concave mirror will directly heat up the propellant tank. The propellant tank is placed in the mirror's focal axis.

The present paper tries to solve the operational life issue by developing a propulsion system for satellites which will increase the operational life of the satellite on the orbit.

Tests at low temperature (specific for cold gas propulsion) and tests at high temperature (specific for hot gas propulsion) will be carried out. A comparison between the forces obtained in both cases for different pressures will be carried out. The aim is to see what improvement in performance will the hot gas propulsion system bring, given the same amount of mass of $N_2$ used in both the cold and hot gas propulsion systems.

The new type of solar–thermal propulsion system developed by COMOTI in the frame of a national space research program STAR—project STRAUSS—can be a solution for the satellite's operational life increase.

## 2. New Type of Solar–Thermal/Cold Gas Propulsion Description

The operating principle of this new propulsion system is based on the focus of solar light. According to this concept, the working life of thrusters can be increased if additional energy is taken from Sun for the propellant's internal energy increase before its expansion in the nozzles.

The new propulsion system is composed of a concave mirror having a small service propellant tank placed along its focal line (Figure 3).

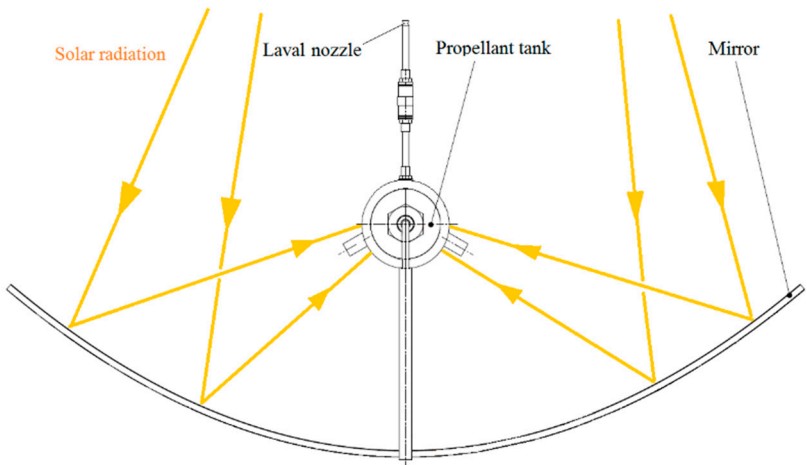

**Figure 3.** Hot gas propulsion system working principle.

The cold gas (nitrogen, $N_2$) is transferred from the main tank with a specific time before expansion. The service tank is a small cylinder made of stainless steel which is painted in black for a total absorption of solar light. When the solar light is focused on the service propellant tank, its temperature increases with several hundred degrees Celsius, thus heating the gas inside.

During expansion, the propulsion force is higher than in the case of a cold gas because the expansion speed is higher. In this way, an important quantity of gas is saved at every gas expansion for attitude correction. The obtained reaction force is sufficient to change the satellite's attitude, so that it maintains its correct position in space. By heating up the propellant inside the service propellant tank, more corrections of attitude/gas expansions than in the case of classic cold gas propulsion system can be achieved.

## 3. Experimental System Description

In order to prove the working principle of the system, experimental equipment was built for laboratory testing. For testing, the radiation source—which in real life is the sun—was replaced with an infrared lamp system which emits an equivalent radiation with the one present on the LEO. The design of the experimental equipment destined for the testing of the propulsion principle is presented in Figure 4.

The testing equipment consisted of a light source (1) which was an infrared lamp system—for generating the parallel infrared rays. The infrared rays were focused by the mirror (2) on the central focal line. The service propellant tank (3) was mounted on the central focal line of the mirror, therefore it was heated up. The cold $N_2$ gas was fed from the main tank (4) which had $N_2$ at 200 bar into the pressure regulator (5). The pressure regulator reduced the gas pressure to 20 bar and transferred the gas with this pressure to the service propellant tank.

When the solenoid (6) valve opened briefly (t = 0.1–0.4 s), a certain a quantity of gas was exhausted through the Laval nozzle (7) generating thrust force. The thrust force was measured by the force transducer (8). Five thermocouples were used for the temperature measurement; one mounted on the propellant tank (9a), three used for the heat dissipation measurement along the distance from

the radiation source to the mirror (9b), and one thermocouple was used for the room temperature measurement (9c).

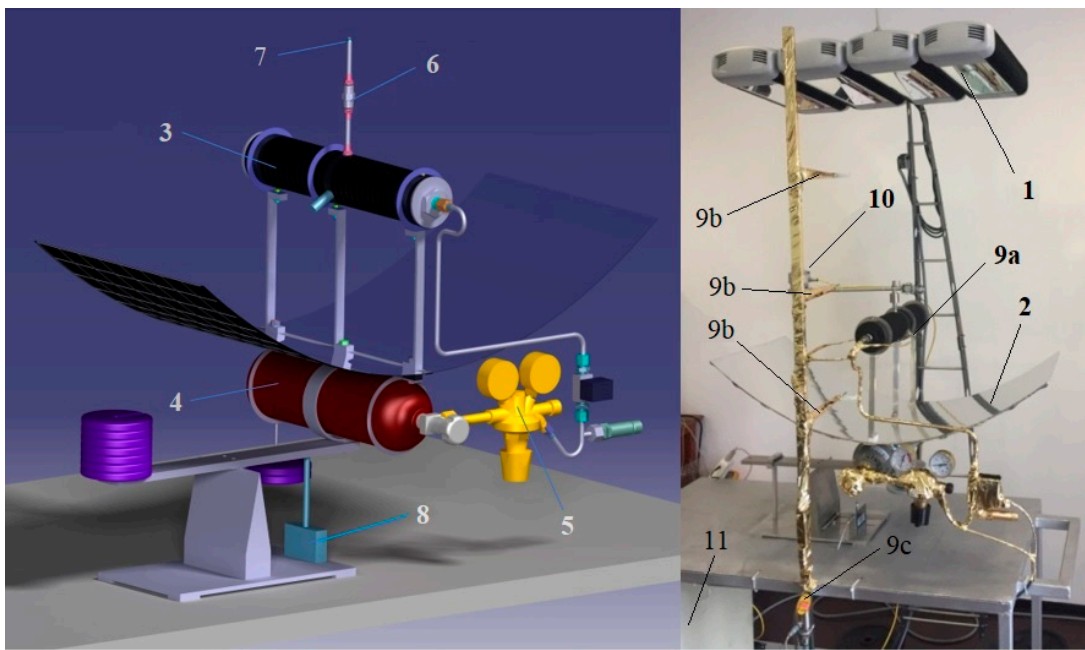

**Figure 4.** The design (**left**) and photo (**right**) of the experimental equipment.

The pressure transducer was mounted on the service propellant tank. For measuring the irradiation, one pyrometer (10) was used.

The values of the gas pressure, temperature and thrust force were transmitted to a data acquisition system (11).

Furthermore, the solar constant at the mean distance of the Earth from the Sun was measured to be S = 1366 W/m$^2$ (with an uncertainty of ±3 W/m$^2$) [27]. Therefore, the radiation power used in the experiments was P$_R$ = 1360 W/m$^2$. The infrared light source is capable of emitting 5000 W/m$^2$, being powered with 220 V at 60 Hz. A potentiometer was used to control the intensity of the light. The radiation source was placed at 1 m above the focal axis of the mirror.

The irradiance value was recorded by the pyranometer. Convection is negligible in space, given the absence of sufficient gas molecules for energy transportation. Therefore, only an approximation for energy loss through radiation (due to mirror heating) can be done. The radiation losses are determined with the Stefan–Boltzmann law [28], see Equations (2) and (3) below in the article.

The solenoid valve functions between −10 °C and 125 °C with air or nitrogen. The shortest open/close time interval is 0.02–0.03 s. It can operate between 0 and 90 bar and is powered with 24 V A.C. current [29].

*3.1. The Mirror*

The main important piece of the equipment is the concave mirror. The mirror calculation was carried out as in [30]. Design and dimensions of the mirror can be seen in Figure 5.

The opening of the mirror/aperture is *L* = 904 mm, mirror width is *l* = 410 mm, and the focal line height is *h* = 268 mm from the mirror base. The parabola parameter is *f* = *p*/2 = 268 mm and its Equation is given by (1):

$$y = \frac{x^2}{4f} = \frac{x^2}{4 \times 268} = \frac{452^2}{4 \times 268} = 190.58 \text{ mm} \tag{1}$$

where $p$ is the parabola parameter which can be expressed as $p = 2f$. The parabola Equation can be expressed as $x^2 = 2py$; $y$ is the vertical Cartesian coordinate in which the parabola is represented, $x$ is the horizontal Cartesian coordinate, $f$ is the ordinate of focus point, $F$. For the parabola: $F(0, f)$ when the parabola Equation is given by Equation: $y = (1/4f)x^2$. Replacing $x$ with half of the aperture and $f$ with the height of the focal line from the mirror base, we obtain the height of the mirror, $y$ as in Figure 6, with the value of 190.5 mm.

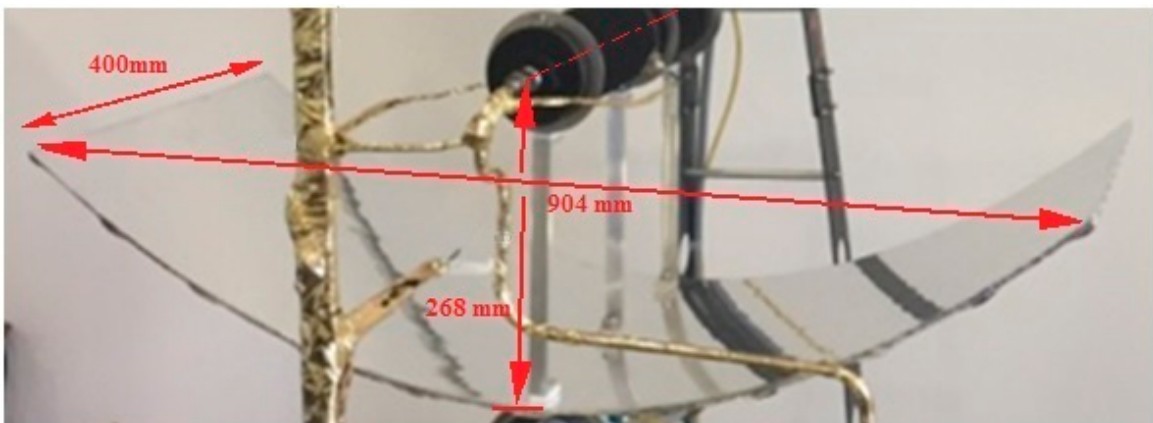

**Figure 5.** The design and photo of the mirror of the experimental equipment.

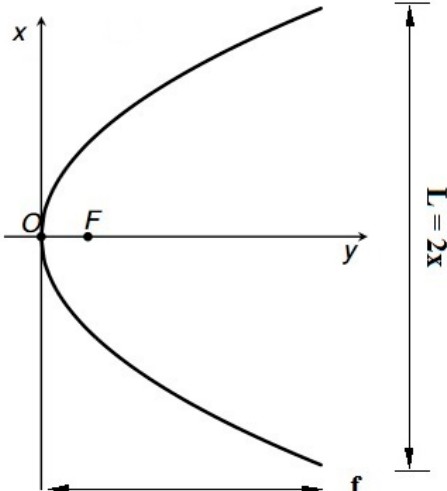

**Figure 6.** Parabola Equation.

The focus point is the converging point of all the reflected rays when the incident rays are in parallel with the OY axis of the coordinates. The mirror is made of carbon fiber-reinforced polymer, CFRP. It has a total thickness of 7 mm and has ribs for stiffness on the opposite side of the mirror's reflective face.

In order to build the concave mirror, a specific mold was required. The mold was made out of cast iron (Figure 7).

It features a fine polished concave side according to Equation (1) and with the same dimensions as presented in Figure 5 for the concave mirror.

Multiple layers of pre-impregnated carbon fiber with a thickness of 0.3 mm were used for the concave mirror manufacturing.

During this process, the cast iron mold was covered with resin-absorbing cloth, placed in a vacuum bag which was vacuumed at $p = 0.01$ bar and treated in oven at 180 °C for 6 h.

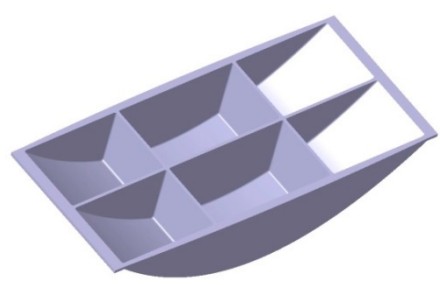
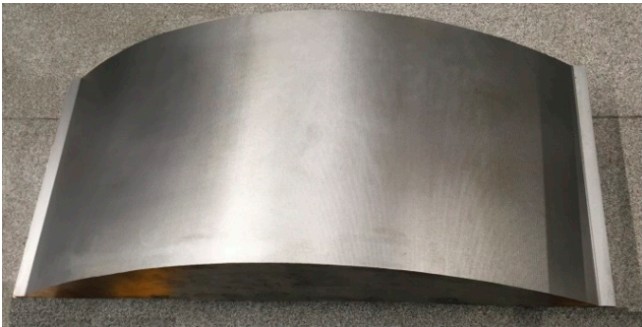

**Figure 7.** The design (**left**) and photo (**right**) of the mold for the forming of the concave mirror.

Finally, a highly reflective aluminum foil was glued on the concave part of the concave mirror. The edges of the reflective surface were additionally secured with thin strips of aluminum tape, ensuring the reliable contact between the reflective foil and mirror.

### 3.2. The Design of the Service Propellant Tank

The service propellant tank has a volume of 1.6 L and houses 462 g of $N_2$ at $p_{tank}$ = 20 bar. The pressure $p_{tank}$ = 20 bar is the $N_2$ pressure inside the buffer tank for low temperature testing. It is also the pressure from which the tank will be heated for the high temperature testing. In the conditions of the experiments conducted at sea-level atmospheric conditions, the propellant tank can reach 125 °C. The tank has an outside diameter of 80 mm, an inside diameter of 50 mm and a length of 400 mm. More design details of the service propellant tank are presented in Figure 8. It features the valve and the Laval nozzle fitted at the middle length, with an additional two holes for the pressure and temperature transducer, which are fitted at 120 °C (from the vertical axis) at either side. The hole placed vertically is used to mount the solenoid valve to the propellant tank. The cap on the left side features a threaded hole through which the piping carrying $N_2$ is fitted.

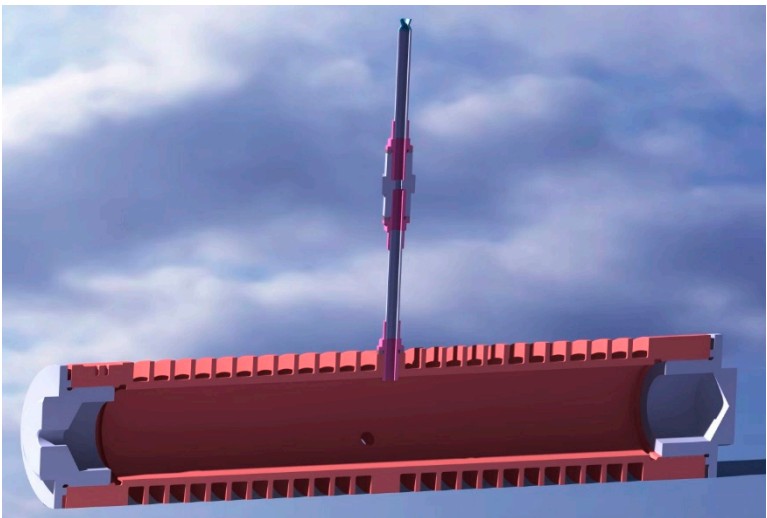

**Figure 8.** D service tank and solenoid valve design.

The service propellant tank was made of 310S-type stainless steel (with a high percentage of Nickel) and features circular stiffening ribs. It has a mass of 6100 g, the end caps included, without the valve-nozzle assembly which have the mass of 450 g.

The propellant tank is painted with a black mate heat resistant paint. It is placed in the focal axis of the concave mirror, thus absorbing the infrared light emitted by the infrared lamps and focused by the concave mirror.

### 3.3. The Laval Nozzle

The design of Laval nozzle [31] and real nozzle can be seen in Figure 9a,b.

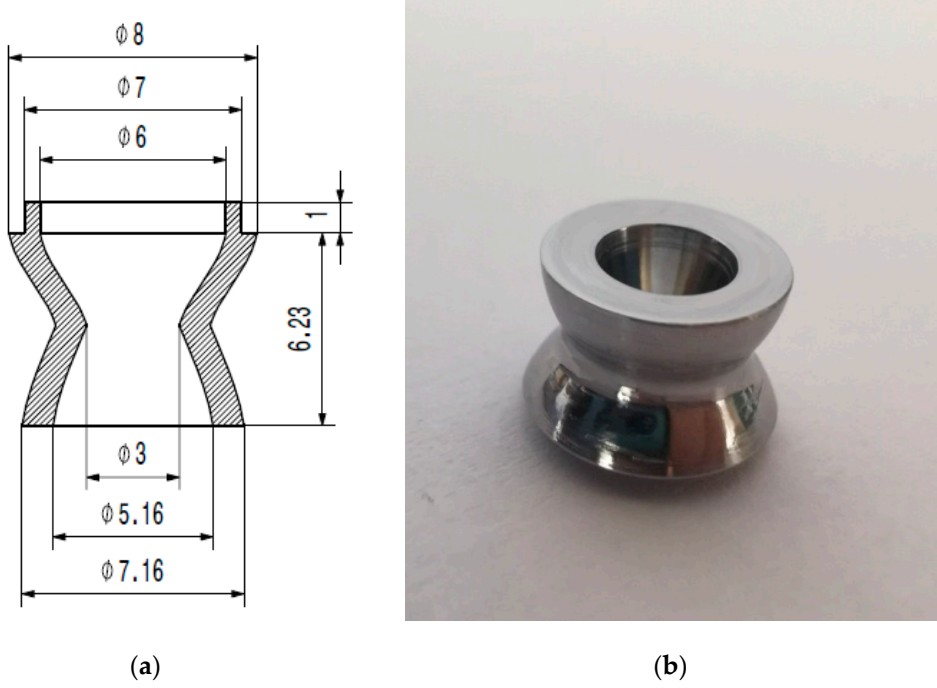

(a)  (b)

**Figure 9.** (**a**) The design of Laval nozzle; and (**b**) Laval nozzle.

The nozzle critical section has a diameter of 3 mm. The nozzle is soldered with silver on a fitting which is screwed into the solenoid valve exhaust end. The valve diameter is 4 mm, less than the fitting inner diameter of 5 mm. The fitting is screwed into the solenoid valve structure right up to the level of the valve. This design ensures the smooth operation of the valve, with a minimal loss of $N_2$.

The Laval nozzle is made out of 316 L stainless steel using a computer numerical control (CNC) milling machine.

The upper structure of the hot gas propulsion system consisted of the propellant tank's supports and ring supports. The lower part of the supports was attached to the mirror, while on the upper part, a total of three supporting rings were fixed on each support. The supporting rings were laser-cut from 316 L stainless steel sheet. Between the propellant tank support and the mirror, polytetrafluoroethylene (PTFE) spacers 5 mm-thick were added to prevent the concentrated heat transfer on the mirror. The ring supports were fixed with the propellant tank supports using M5 screws, nuts and washers. The propellant tank was supported by the three supporting rings. The lower support system was attached to the underside of the mirror with the screws that passed through the upper support system, Teflon spacers and mirror. The lower support system consisted of a central "C" shaped support and two lower support rings for the main feed tank. The central C-shaped support had the mirror fitted on the upper part and its lower part was fitted on the balance arm end. The central support system was the only element which connected the mirror and its upper elements to the balance arm. The lower support rings were laser cut from a 316 L stainless steel sheet. The main propellant tank was inserted into the two rings and central support system. The support ring which fixes the rear of the main tank has a balance pan attached to it. It was used to put weights on it in order to cancel out the imbalance due to weight of the piping, pressure regulator, safety valve and electromagnetic valve mounted on the front part of the system, as shown in Figure 10.

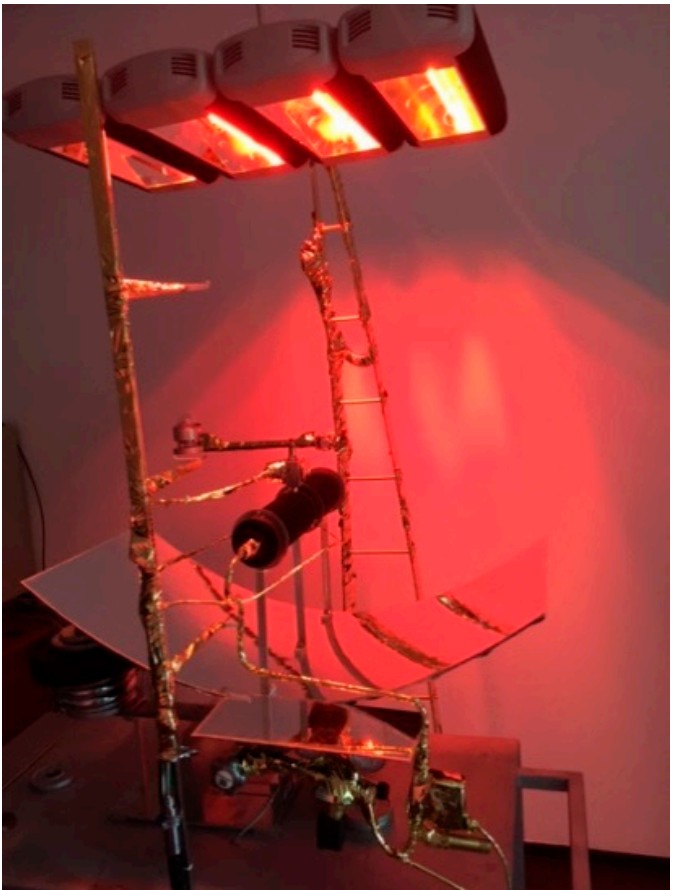

**Figure 10.** Testing equipment during experiments.

When the infrared lamps were activated, they radiated parallel rays of infrared light to the concave mirror which concentrated them on its focal line where the service tank was placed.

In order to avoid force transducer damage during the experiment, the balance lever position was limited by a position limiter (Figure 4), which was fixed on the base plate. The position limiter does not allow the tilting of the balance lever beyond a certain value. The travel limit of the balance lever is controlled by a thin pitch screw which pushes on the balance arm on its upper side.

The force transducer (Figure 4) was a high precision one, measuring a maximum force of 50 N with a precision of ±0.01 N. The force transducer also had the option to record 10 readings and the PC live monitoring of measurements.

The force transducer was mounted between the base plate and the balance lever (Figure 4). It featured a guide path machined into the base plate, which allowed for the transducer to slide across the length of the balance arm. The guide channel had graduations of 1 mm across its length. The center of the force transducer was marked, thus enabling for the easy read of the length of the arm at which the force was applied by the hot gas propulsion system.

In Figure 10, one can see that the pyranometer is placed at a distance of 1 m from the radiation sources and at 200 mm above the focal axis of the mirror. It measures the temperature (°C) and irradiance (W) emitted by the radiation sources when the arm is being hinged into the path of the infrared light from the sources.

To further measure how much of the infrared light reaches the mirror and how much the mirror reflects it back, the three thermocouples for the heat dissipation measurement along the distance from the radiation source to the mirror are fitted at a height of 1.3, 1.6, and 1.9 m from the radiation source. The elements exposed to the infrared light were covered in gold-plated foil: the radiation sources

support, the pyranometer's hinged arm, the three thermocouples supports and cables, and the cable from the force transducer.

## 4. Experiments and Results

The functionality and principle check of the experimental system was carried out through experiments in two stages, part A and part B.

Part A. The first stage in the test campaign comprised performing the tests without the energy source (radiation) concentrated by the mirror cold testing in which the system was a cold gas propulsion system.

Part B. The second stage comprised the hot gas testing in which the radiation source which simulates the sun was activated and its energy was concentrated with the mirror.

Due to the fact that the measurements were done in atmosphere, the maximum real temperature of the service tank which was achieved was 125 °C, the maximum allowed for the operation of the solenoid valve. For this reason, the real case was only simulated for the temperature of 450 °C. The temperature of 450 °C is the maximum temperature that can be obtained using the available surface of the mirror, considering that the sun rays are uniformly concentrated on the actual surface of the buffer tank. According to the mentioned hypothesis, the absence of radiation is added, so that by applying the Stefan–Boltzmann law, the mentioned value is obtained:

$$\varnothing = \varepsilon \sigma T^4 \implies T = \sqrt[4]{\frac{I}{\sigma} \frac{S_{mirror}}{S_{tank}}} = 469 \,°C \tag{2}$$

where $\varnothing$ represents the radiant flux, $I$ = 1361 W/m$^2$—solar irradiance, Stefan-Boltsmann constant $\sigma = 5.67 \times 10^{(-8)}$ W/m$^2$ K$^{(-4)}$, $\varepsilon$—emissivity, mirror surface $S_{mirror}$ = 0.2527 m$^2$, buffer tank surface $S_{tank}$ = 0.1465 m$^2$ (corresponding to $\Phi$ = 80 mm and a length of 400 mm). Decreasing the tank surface area (to $\Phi$ = 20 mm, hence a total surface of 0.0253 m$^2$) will lead to an ideal temperature of 696 °C (without radiation losses). Radiation losses $q$ result according to the heat radiation law:

$$q = \varepsilon \sigma \left( T_h{}^4 - T_c{}^4 \right) S_{tank} \tag{3}$$

where $T_h$—hot body absolute temperature, and $T_c$—cold surroundings absolute temperature.

### 4.1. Part A—Cold Gas Propulsion

The testing and validation of the principle of solar–thermal cold gas propulsion were carried out in logical stages, in order to ultimately obtain a conclusive result. It was noticed from the first tests that the opening time of the solenoid valve was a crucial factor for the obtained value of the mean force obtained in a given $\Delta t$ and for the total impulse obtained, utilizing a well determined N$_2$ quantity. In this regard, multiple experiments were carried out at low temperature for which the impulses for more opening times of the valve were measured. A fixed quantity of N$_2$ was used (which corresponds to the fill up of the buffer tank at 20 bar). The measured impulses were later summed up in order to make a quantitative comparison. The results for such a test are presented in Table 1.

For the solenoid valve opening times of 0.1 and 0.3 s, the values were analyzed below.

The initial data revealed the fact that there was an error for the force measurement, therefore the calibration of the force transducer was set to start at the value of 10 N and not 0 N.

In this regard, it was noticed that there was no major difference between the total impulse obtained with the different opening times. Therefore, the tests were oriented at small valve opening times such that the average force obtained would be in the value vicinity of 1 N, common force values for satellite attitude correction.

As it can be noticed in Figure 11, an exponential drop of force was obtained when using cold gas propulsion.

**Table 1.** Total impulse at cold temperature, at several valve opening times.

| Opening Time (s) | | | | | | | |
|---|---|---|---|---|---|---|---|
| 0.2 | | 0.4 | | 0.5 | | 0.6 | |
| Isolated Impulse (Ns) | Average Force per Impulse (N) | Isolated Impulse (Ns) | Average Force per Impulse (N) | Isolated Impulse (Ns) | Average Force per Impulse (N) | Isolated Impulse (Ns) | Average Force per Impulse (N) |
| 2.072 | 2.309 | 3.819 | 2.546 | 4.351 | 2.901 | 4.75 | 2.969 |
| 1.555 | 1.728 | 2.104 | 1.365 | 1.837 | 1.67 | 1.654 | 1.504 |
| 1.163 | 1.292 | 1.293 | 1.288 | 0.672 | 0.747 | 0.828 | 0.753 |
| 0.721 | 0.811 | 0.641 | 0.712 | 0.402 | 0.447 | 0.257 | 0.321 |
| 0.487 | 0.541 | 0.347 | 0.434 | | | | |
| 0.483 | 0.604 | 0.141 | 0.176 | | | | |
| 0.333 | 0.416 | | | | | | |
| 0.242 | 0.303 | | | | | | |
| 0.196 | 0.28 | | | | | | |
| 0.14 | 0.2 | | | | | | |
| 0.071 | 0.101 | | | | | | |
| **Total Impulse Value (Ns)** | | | | | | | |
| 7.463 | | 8.345 | | 7.262 | | 7.489 | |

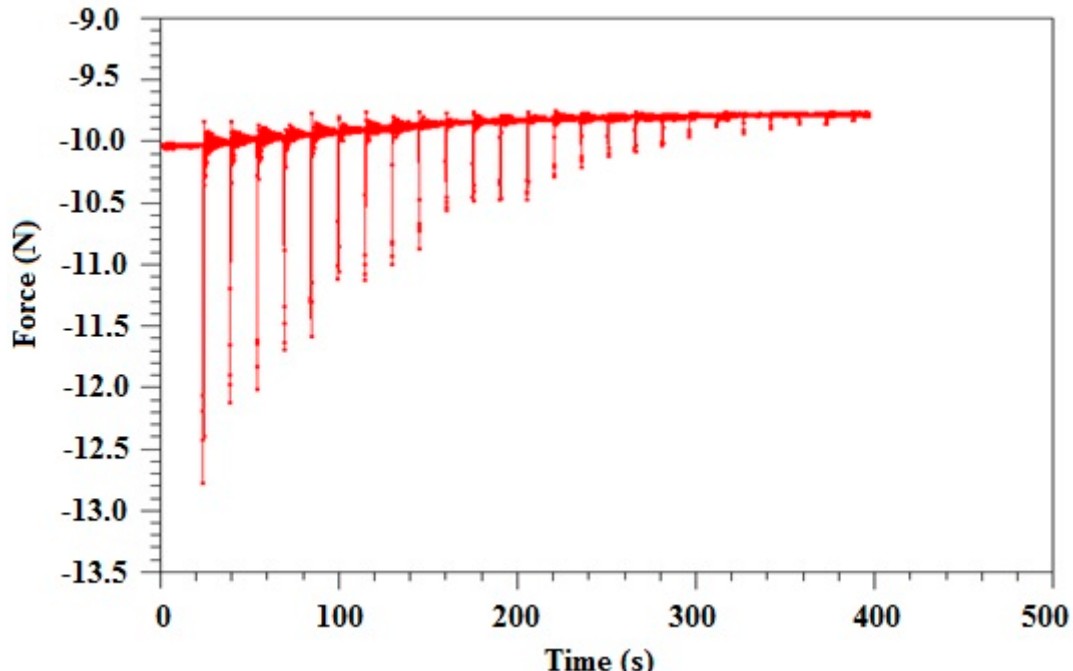

**Figure 11.** Force variation with time at the valve opening time of 0.1 at a 10 s interval.

Because the system implies the occurrence of sudden variations (impulses—mechanical in the solenoid valve and gasodynamic in the atmosphere), an oscillatory movement of low amplitude occurs, but noticeable by the data acquisition system. This oscillatory movement tones down, corresponding to the 10–15 s interval between the opening and closing of the valve. Moreover, it can be observed

from the graphical representation that the mass of discharged fluid from the system can be determined from the difference of weight (at the beginning and at the end of the experiments).

### 4.2. Part B—Hot Gas Propulsion

In order to test the functionality of the test equipment and the principle, a test at 80 °C was carried out, the temperature being within the operation limits of the valve and of the entire test equipment. After this test, a comparison between the impulses at cold gas and at hot gas was made.

The illustration of the impulse of the cold gas (from Figure 11) in parallel with the impulse from the hot gas are represented in Figure 12. The tests at low temperature are displayed in blue and the one at high temperature, 80 °C, in red.

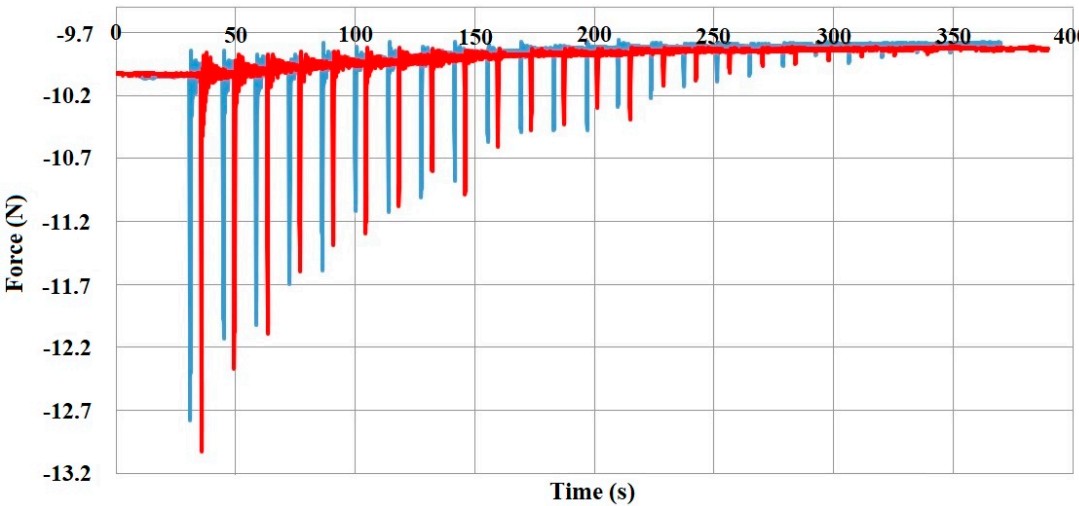

**Figure 12.** Comparison between the impulse for cold gas ($T_0$ = 26 °C) and hot gas ($T_f$ = 80 °C) for the valve opening time of 0.1 s at a 10 s interval.

It can be noticed that the force value for the hot gas is somewhat bigger than the force value obtained for the cold gas, in total, as a significant increase in total impulse is recorded. The impulse increase is in absolute values from 7.87 to 8.02 Ns, with an increase of 2%. This difference is percentage wise so small because at small values of opening time, there is a strong influence coming from the valve recoil. Therefore, at 0.05–0.1 s, a force of small intensity can be noticed which acts in opposition to the movement, but which destroys the force distribution. This influence is present utilizing the detail from Figure 13, corresponding to the first impulse from the cold gas from Figure 11.

The recoil effect is represented in Figures 13 and 14b. The recoil effect reduces the thrust force on average by 0.3 N, irrespective of the valve opening time. This is due to the direction of the valve movement, in opposition to the direction of the thrust force. In Figures 13 and 14b, the force would have been represented by a spike in the line, in the opposite direction from the spike which is represented in both figures. Therefore, the developed force of the discharged hot gas would have been higher by 0.3 N.

If the case with a valve opening of 0.3 s is analyzed, the valve influence is less pronounced, therefore, there is an increase of 2.4% for the same temperature interval.

As an evolution, it was observed in Figure 14a that in this case, there is also a difference between the force and the impulse for the cold gas in comparison with the hot gas. The influence of the valve recoil on the force/impulse can be seen in Figure 14b.

The fact that the gas warming inside the tank was carried out with respect to the state Equation must be noticed. Therefore, in Figure 15, the pressure variation with temperature increase xis presented. This variation is displayed in Figure 13 for this case, together with the relevant detail in the warming

interval. The warming interval is 26–80 °C. The decrease in pressure after 80 °C corresponds to the valve opening in order to produce the impulses.

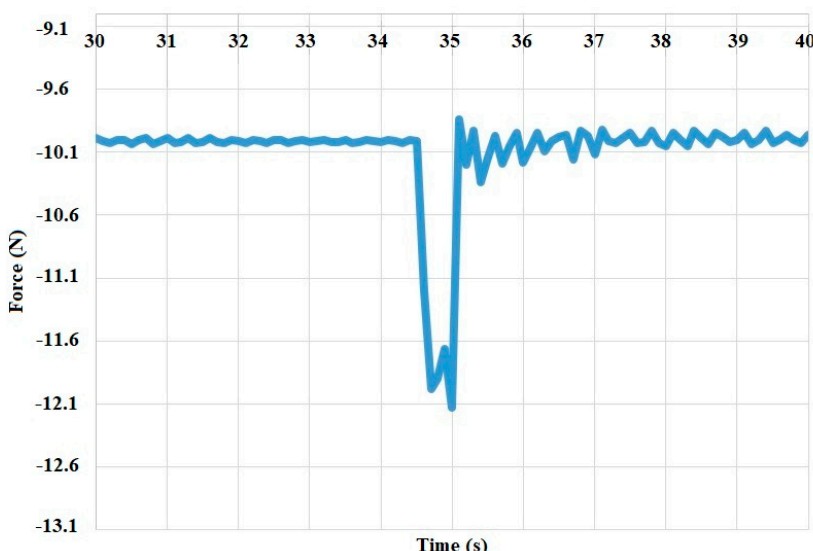

**Figure 13.** The influence of the valve recoil on the impulse.

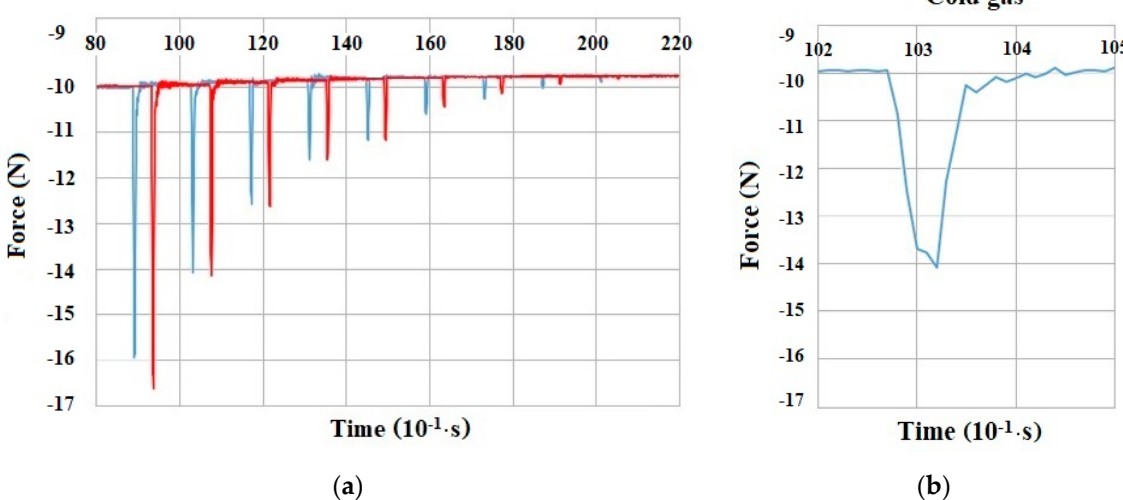

(a)                                                              (b)

**Figure 14.** (**a**) Impulses for the hot and cold gas for $\Delta t$ = 0.3 s and warming from 26 °C to 80 °C; and (**b**) cold gas force for $\Delta t$ = 0.3 fs and valve recoil.

The following fact must not be neglected: in the moment of a $N_2$ quantity discharge from the service tank, the temperature drops suddenly (around 3 °C) because of the mass ejection outside the system.

The fluid temperature in the system tends to increase because of the tank thermal inertia, the nitrogen warming being more pronounced as more quantity of fluid from the tank is reduced with each valve opening. Furthermore, as the quantity reduces with each discharge in the atmosphere, the temperature inside the tank does not decrease as much as at in case of the first discharge. This can be noticed in the 75–85 °C temperature interval in Figure 15, where at 80 °C the valve opens.

Because the correction maneuvers on the orbit require (usually) equal impulses, the comparison of the cold gas impulses with the impulses after the service tank heat-up from 22 °C to 60 °C–100 °C–125 °C was carried out. The upper temperature limitation was due to the solenoid valve material sensitivity, which was 125 °C. As regards the experiment, of special interest was the total impulse obtained at

different initial temperatures of the buffer tank, at a valve opening time of 0.4 s. This opening time was determined as optimal for a 50 bar total pressure drop from the main tank (which corresponds to a 114 g discharged mass). By integrating the force related to time, in the necessary interval of 50 bar, a total impulse was obtained in accordance with Table 2.

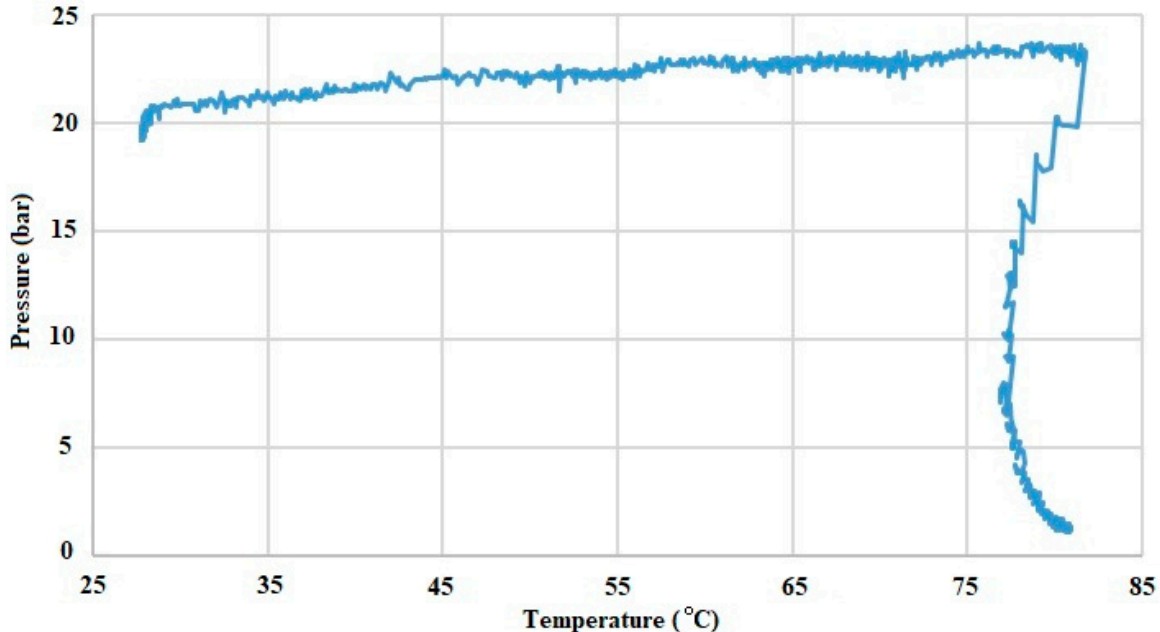

**Figure 15.** Pressure variation with the temperature warming from 26 °C to 80 °C (experiment from Figure 12).

**Table 2.** Total impulse value at $\Delta t = 0.4$ s and different temperatures of the buffer tank.

| Temperature (°C) | Total Impulse (Ns) | Related Impulse (Ns/bar) |
|---|---|---|
| 22 | 63.559 | 1.27118 |
| 60 | 70.37 | 1.4074 |
| 100 | 83.643 | 1.67286 |
| 125 | 89.105 | 1.7821 |

Through extrapolating these results for the entire temperature interval, using the functions from Figure 16, a significant percentage increase (Table 3) can be obtained. Two laws of extrapolation were considered, one linear and the other one polynomial of second degree.

Usually, the pressure and temperature vary proportionally, with respect to the ideal gas law, by taking into consideration no gas loss from the tank. The experimental data present a semi-linear pressure evolution with respect to the temperature in the 25–125 °C interval. The evolution will not be 100% linear given the fact that heat will be lost in space through radiation. The radiation loss is not linear, it varies with $T^4$, where T is the tank temperature. If the radiation is not taken into account, the evolution is linear. If it is desired to take the radiation into consideration, it must be determined how the p-T variation drifts away from the curve. A balancing between the incoming radiation flux from the sun, the radiation given to the tank and the heat lost through the radiation must be made.

The data extrapolation is presented in Figure 16.

It can be noticed that two extrapolation types were taken into account: a linear extrapolation, which is closer to the intuited value from the start of the project (a 2.5× increase in the total impulse at maximum temperature) and a polynomial extrapolation. It is impossible to specify an exact value due to the experimental conditions given by the solenoid valve. The fact that a potential 2.7× growth can

be obtained at a temperature of 450 °C (temperature which can be reached at Earth's low orbit) is not to be neglected. The polynomial extrapolation is more optimistic (it approximates a 640% increase), although more precise results can be obtained using a more dedicated valve.

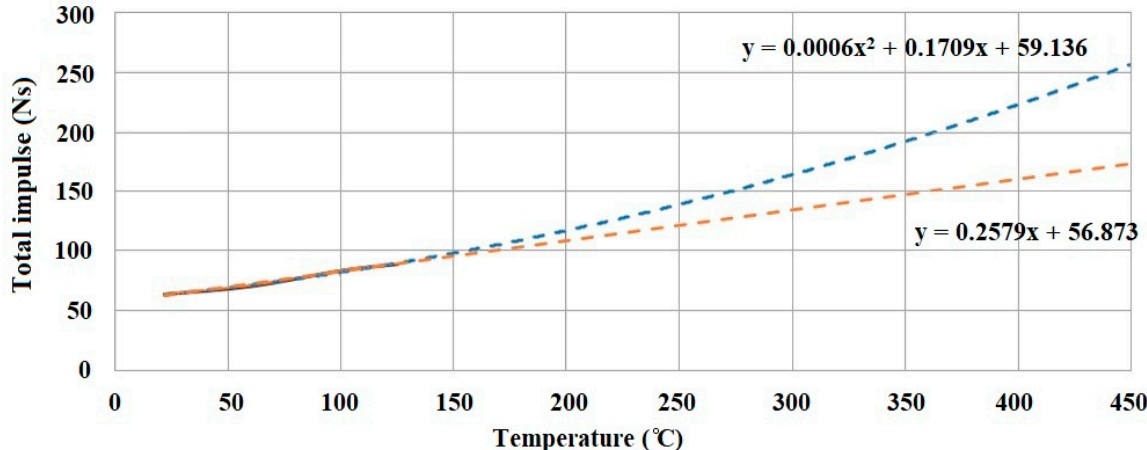

**Figure 16.** Experimentally obtained impulse using $N_2$ at 50 bar and extrapolation from 125 °C to 450 °C at $\Delta t = 0.4$ s and different buffer tank temperatures.

**Table 3.** Total impulse value at $\Delta t = 0.4$ s and different temperatures of the buffer tank (extrapolation).

| Temperature (°C) | Impulse (Poly) (Ns) | Related Impulse (Ns/bar) | Percentage Growth (Polynomial) | Impulse (Linear) (Ns) | Related Impulse (Ns/bar) | Percentage Growth (Linear) |
|---|---|---|---|---|---|---|
| | | | Tested values | | | |
| 25 | 63.92 | 1.27836 | 0.00 | 63.31 | 1.26623 | 0.00 |
| 50 | 68.14 | 1.36276 | 6.60 | 69.76 | 1.39518 | 10.18 |
| 75 | 74.86 | 1.49716 | 17.12 | 76.21 | 1.52413 | 20.37 |
| 100 | 84.08 | 1.68156 | 31.54 | 82.65 | 1.65308 | 30.55 |
| 125 | 95.80 | 1.91596 | 49.88 | 89.10 | 1.78203 | 40.74 |
| | | | Predicted values | | | |
| 150 | 110.02 | 2.20036 | 72.12 | 95.55 | 1.91098 | 50.92 |
| 175 | 126.74 | 2.53476 | 98.28 | 102.00 | 2.03993 | 61.10 |
| 200 | 145.96 | 2.91916 | 128.35 | 108.44 | 2.16888 | 71.29 |
| 225 | 167.68 | 3.35356 | 162.33 | 114.89 | 2.29783 | 81.47 |
| 250 | 191.90 | 3.83796 | 200.23 | 121.34 | 2.42678 | 91.65 |
| 275 | 218.62 | 4.37236 | 242.03 | 127.79 | 2.55573 | 101.84 |
| 300 | 247.84 | 4.95676 | 287.74 | 134.23 | 2.68468 | 112.02 |
| 325 | 279.56 | 5.59116 | 337.37 | 140.68 | 2.81363 | 122.21 |
| 350 | 313.78 | 6.27556 | 390.91 | 147.13 | 2.94258 | 132.39 |
| 375 | 350.50 | 7.00996 | 448.36 | 153.58 | 3.07153 | 142.57 |
| 400 | 389.72 | 7.79436 | 509.72 | 160.02 | 3.20048 | 152.76 |
| 425 | 431.44 | 8.62876 | 574.99 | 166.47 | 3.32943 | 162.94 |
| 450 | 475.66 | 9.51316 | 644.17 | 172.92 | 3.45838 | 173.12 |

Given the mission of the satellite, the propellant mass onboard can vary from satellite to satellite. As a comparison value, the propellant mass is 1/5 the mass of the nano-satellite Adelis-SAMSON,

which is fitted with a cold gas propulsion system. Adelis-SAMSON carries 468 gr of Krypton at 160 bar and 30 °C. This ensures a total impulse of >150 Ns [32]. In comparison, the hot gas propulsion system develops a total impulse of 172 Ns with nitrogen at 450 °C and 27 bar, heated from 25 °C and 20 bar. Despite the fact that direct comparisons are difficult, as the tank volume and pressure are different for both solutions, it can be noticed that the hot gas propulsion system offers a 14.6% increase in the total impulse, with same mass of propellant but at six times-lower propellant pressure.

## 5. Conclusions

The conclusion of the tests and the subsequent data extrapolations is that the specific impulse of the hot gas propulsion system increases by 173.24% over the obtained specific impulse at 25 °C where no heating of the tank was carried out. The temperature of 450 °C is the temperature which the propellant tank reaches in space, when its heating is carried out by the mirror. Therefore, the operational life of an LEO satellite increases 2.73 times. The maximum thrust force is 8 N when the valve opening time is 0.4 s.

The maximum thrust force which the hot gas propulsion system achieves is 9.21 N, at 27 bar, achieved after heating up the buffer tank to 125 °C and opening the valve for 0.3 s.

The limitations of this study are measurement related. The discharged gas speed at the nozzle was neither measured or determined. Therefore, the specific impulse could not be calculated and as a result, the total impulse (which is correlated with the autonomy of the hot gas propulsion system) was not determined from the specific impulse. By determining the total impulse from the specific impulse, less result errors would have occurred and the calculation procedure would have been more straightforward.

Therefore, for future experiments, a lightweight propellant tank will be developed with a 0.7 L volume and fitted with six valves and nozzles

In the future, it is of interest to test the new concept for a wider range of temperatures, up to 450 °C, which is the mirror temperature threshold. It is intended to carry out the tests in vacuum and temperature conditions similarly to the values from the cosmic space for LEO.

Important aspects which will arise when deploying the LEO satellite onto the orbit are control related. The linear parabolic mirror destined for space will comprise two symmetrical parts which will be folded until the first correction maneuver. When they unfold, they will create the complete mirror. The mirror will be silver plated and the valve-nozzle assemblies will be fitted in a manner to cover all ranges of necessary correction movements, which means a 360° action. Problems that will arise are control-related: The correction maneuver can alter the position of the satellite in space, therefore, all corrections will be limited to a valve opening time of maximum 0.3 s. Therefore, no gas will be wasted. The mirror destined for testing has a mass of 2105 g. The mirror destined for the space application will have a mass of 1100 g and will be made out of 2 mm-thick CFRP. Furthermore, the service tank/buffer tank will have a mass of 460 g. Including all the valves, piping and sensors, the total mass of the system destined for space will be of 3150 g, therefore 1150 g heavier than a comparable cold gas propulsion system, in terms of tank capacity, quantity of propellant carried onboard and developed thrust.

**Author Contributions:** Conceptualization, C.S. and G.C.; methodology, H.Ș.; software, A.R. and A.T.; validation, A.T.; writing—original draft preparation, H.Ș., C.S., G.C., A.T., T.T., A.R.; writing—review and editing, H.Ș., G.C.; supervision V.S.; project administration, V.S. All authors have read and agreed to the published version of the manuscript.

**Funding:** This work was carried out within STAR, "STRAUSS" Project, Grant no. 130/2017, Project supported by the Romanian Minister of Research and Innovation.

**Conflicts of Interest:** The authors declare no conflict of interest.

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
