# Peer review of "On a New Type of Combined Solar–Thermal/Cold Gas Propulsion System Used for LEO Satellite’s Attitude Control"

_applsci, doi:10.3390/app10207197_

Round 1

Reviewer 1 Report

I am sorry to inform the authors that I have decided my opinion as reject for this article according to the following reasons.

1.  English should be improved. Sentences are lengthy and some sentences fail to deliver a meaning of what they have to do. Also logical flow of statement are poorly organized, and structure of paragraph  (lines and spacing) as either.

2. There are number of typos in the article. (ex. line 111, line 137, line 257, line 395)

3.  The quality of fig. 2 should be improved.

4. There is no specific data of that infrared lamp can replace the radiation source (i.e. sun) simulating in LEO. Numbers and data should provided.

5. Fig. 7a and fig. 7b looks like the same ones. There are no differences.

6. For designing the radiation test, an effect of convection must be stated, yet missing in this article.

7. In series of statement for experimental equipment, irrelevant information are too much given (ex. machining method), yet missing key numbers of specifications.

8. Temperature limit of test (125 Celsius) is too low to extend the predictable findings in practical environment of 450 Celsius. 

9. The authors suggested the performance in practical environment by employing polynomial and linear models. The reviewer believe that the authors should suggest a reasonable model based on thermodynamics, not just curve fittings.

Author Response

Above are the detailed corrections we made according to your comments point by point.

Thanks again for your meticulous review and valuable suggestions to improve our manuscript. We hope that the revision has addressed all the issues in the old version. However, in the event that the modifications listed above DO NOT meet your expectations, we sincerely request you to give us another opportunity to revise the paper. We are very grateful for this opportunity given to us and would be more than happy to revise the paper until you are satisfied. We are looking forward to your positive response. Thank you again and kudos to you!

Best regards,

Constantin Sandu, Valentin Silivestru, Grigore Cican*, Horațiu Șerbescu, Traian Tipa, Andrei Totu, Andrei Radu

Reviewer 2 Report

Review on the manuscript “On a new type of combined solar-thermal/cold gas propulsion system used for LEO satellite’s attitude control” by Sandu et al., manuscript ID: 950785

In the current manuscript the authors presented the development, construction and testing of a new propulsion system for low earth orbit which combines cold gas propulsion with solar-thermal technology. The paper quite interesting but several issues should be taken in account before publication. Please consider my following suggestions:

  • Define LEO. First time you use an acronym in the text you should give its full definition so then readers will understand the meaning of that acronym. This applies also to the acronym ESA.
  • In the second paragraph of the Introduction authors say: “ESA’s Space Debris Office estimates about 5500 of these satellites will still be in space until February 2020, while 2300 are still functioning [1], most of which are LEO satellites.” Something is wrong in this sentence, February 2020 is past but you are using a future tense…
  • In the sentence starting on line 39 when you are enumerating the different types of propulsion you enumerate two times the cold gas propulsion systems, one at the beginning and one at the end of the sentence.
  • In line 163 you explain that parabola parameter is given by f=p/2. What is f? what is p? define these parameters. The same with equation (1) what is y and what is x?
  • Line 183 what is the ptank? Is it the maximum pressure that the service propellant tank may storage the gas? Please specify it in the text.
  • What is the difference between figure 7a and 7b? Besides, to enumerate them as figure 7a and 7b, they should be placed as two subfigures of figure 7 and only one caption should be used including the caption for subfigures a) and b). Also, figure 7b is not mentioned in the text so it is hard to understand what is the aim of this subfigure.
  • In equation 2 please specify what represents each variable. Define ф ; ε; σ; etc. The same for equation 3.
  • In figure 13 you have the caption for figure 13a but not for figure 13b, please add it.
  • In figure 14 we see that the pressure increases with the increase of temperature but when it reaches 80ºC the pressure decreases with small increases of temperature. Does the initial phase corresponds to the warming phase and when the pressure starts decreasing means that you have started to open the valve in order to produce the impulses? If I am correct, these two phases should be identified in figure 14. If I am not correct please explain it better.
  • The paper should be proofread some typos can be found along the text.

Author Response

(The authors gave the same response as above.)

Reviewer 3 Report

1) Reference [9] is completely missing in the paper text (it exists in the References list).

2) Calls on the figures should be unified through the paper text. The Authors should decide will the call on, for example, Figure 1 will be – Figure 1 or only Fig. 1. It should be applied for all the figures.

3) Line 145 – “N¬2” should be replaced with “N2” (2 in the subscript). Please, check and correct such obvious mistakes throughout the paper text.

4) Titles on Line 157, Line 182 and Line 202 should be sub-sections (of section 3).

5) Line 173 – it is stated: “with the same dimensions as presented in the fig. 3”. In Figure 3 are not presented any dimensions – so the dimensions should be added.

6) Figure 7a and Figure 7b are the same – one of them should be removed. Service propellant tank should be better described and more details should be presented. Characteristics and specifications of solenoid valve are not mentioned at all – this element should also be added and described in the paper.

7) Lines from 274 up to 277 – in the paper text should be used subscripts and superscripts, not ^ or similar non-standard markings.

8) Titles on Line 283 and Line 310 should be sub-sections (of section 4).

9) Figure 11 and Figure 11 title should be placed on the same page.

10) Line 358 – instead “dt”, it should be used delta t as elsewhere throughout the paper before this line. The same should be corrected in Line 369 and Line 373.

11) Table 3 and Table 3 title should be on the same page.

12) Table 3 – what are extrapolation errors (at least approximated or expected errors) for both polynomial and linear extrapolations? Comment this fact and add it into the paper text.

13) Conclusions – is Section 5, not an independent title.

14) Conclusions section must be much better performed. It should be given a wider results and conclusions of the performed research.

15) English through the paper text is ok, however, some sentences can be shortened and better performed. Please, check English throughout the paper text and perform modifications.

16) Solar concentrating system will bring additional mass to LEO satellite. Will this fact be a problem in exploitation? Also, the Authors should discuss in the paper what will be a practical problems and challenges in exploitation. All the benefits of the presented propulsion system and the laboratory tests are well described, but the potential problems and challenges, especially during the exploitation, should be discussed and explained in detail.

17) References should be arranged according to Applied Science instructions for the Authors. Also, several other elements (mostly fonts, alignments, blank lines between the paper text and table titles, etc.) in the paper should be corrected according to Applied Science instructions for the Authors.

Final remarks: This is novel and interesting paper with an important laboratory test results. Keep up the good work in the laboratory. However, before publication this paper requires deep and proper revision, according to my comments above.

Author Response

(The authors gave the same response as above.)

Round 2

Reviewer 1 Report

Thank you for Authors' response, and followings are my comments for this article.

  1. In Fig. 2, there are two identical pictures. Please remove one.
  2. In Fig. 5, the focal height is marked as 265 mm, yet in the paragraph, it is stated as 268 mm (line 181). Please correct.
  3. Kindly recommend to mark the effect of recoil of solenoid valve in Fig. 13 and 14.
  4. In line 400, Figure 14 may be a typo of Figure 15.
  5. In Table 3, please address the tested values and predicted values by adding note column, not just highlighting.
  6. In Conclusion, please state limitations of this study, not just mentioning favorable outcomes.

Author Response

Point 1: In Fig. 2, there are two identical pictures. Please remove one.

Response 1: The new figure number 2 has a better quality .

Point 2: In Fig. 5, the focal height is marked as 265 mm, yet in the paragraph, it is stated as 268 mm (line 181). Please correct.

Response 2: The focal height was correct to 268 mm in Figure 5.

Point 3: Kindly recommend to mark the effect of recoil of solenoid valve in Fig. 13 and 14.

Response 3: The recoil effect is represented in Figure 13 and 14b. The recoil effect reduces the thrust force on average by 0.3N, irrespective of the valve opening time. This is due to the direction of the valve movement, in opposition to the direction of the thrust force. In Figure 13 and 14 b, the force would have been represented by a spike in the line, in opposite direction from the spike which is represented in both figures. Therefore, the developed force of the discharged hot gas would have been higher by 0.3 N.

Point 4: In line 400, Figure 14 may be a typo of Figure 15.

Response 4: Figure 14 was corrected in line 400 with Figure 15.

Point 5: In Table 3, please address the tested values and predicted values by adding note column, not just highlighting.  

Response 5: Tabele 3 was modified

Point 6: In Conclusion, please state limitations of this study, not just mentioning favorable outcomes.

Response 6: The limitations of this study are measurement related. The discharged gas speed at the nozzle was not measured nor determined. Therefore, the specific impulse could not be calculated and as a result, the total impulse (which is correlated with the autonomy of the hot gas propulsion system) was not determined from the specific impulse. By determining the total impulse from the specific impulse, less result errors would have occurred and the calculation procedure would have been more straightforward.

Best regards,

Constantin Sandu, Valentin Silivestru, Grigore Cican*, Horațiu Șerbescu, Traian Tipa, Andrei Totu, Andrei Radu

Reviewer 3 Report

All the required corrections were performed. All the required additional explanations were added. I have no more concerns about this paper. My congratulations to the Authors.

Author Response

Thanks you for these appreciations.

Best regards,

Constantin Sandu, Valentin Silivestru, Grigore Cican*, Horațiu Șerbescu, Traian Tipa, Andrei Totu, Andrei Radu
